# Production of Activated Carbon from Sifted Coke and Determination of Its Physicochemical Characteristics

**DOI:** 10.3390/molecules28155661

**Published:** 2023-07-26

**Authors:** Aigul T. Ordabaeva, Zainulla M. Muldakhmetov, Arstan M. Gazaliev, Sergey V. Kim, Zhazira S. Shaikenova, Mazhit G. Meiramov

**Affiliations:** Institute of Organic Synthesis and Chemistry of Coal of Kazakhstan Republic, Alikhanov Str., 1, Karaganda 100012, Kazakhstan

**Keywords:** carbon sorbent, activation, coke fines, methylene blue, phenol

## Abstract

The possibility of obtaining effective coal sorbents from a low-liquid product of coke chemical production—coke fines—has been studied. To obtain a coal sorbent, coke fines with a size of ≤10 mm were crushed and sieved to obtain a fraction of 2–5 mm. The resulting fraction was activated in a specially designed reactor at 850 °C with steam treatment. Activation was carried out at different durations of the process: 60, 90, and 120 min. It was found that the sample obtained with a process duration of 120 min has the best indicator for the ability to remove phenol from aqueous solutions (74.94 mg/g) and methylene blue (145 mg/g). When cleaning tap water with the resulting carbon sorbent, there is a decrease in the content of calcium, sulfates, and bicarbonate and a decrease in total mineralization. Obtained activated carbon was studied by scanning electron microscopy, low-temperature nitrogen adsorption (BET), and FTIR spectroscopy. It was found that the resulting activated carbon has a porous structure consisting of meso- and macropores, and the specific surface value was ~301 m^2^/g. The presence of high-intensity absorption bands corresponding to acid functional groups has also been established.

## 1. Introduction

Adsorption methods of water purification are attracting more and more attention due to their high efficiency and versatility. They provide effective purification of water from various types of pollutants in a wide range of concentrations. The development of highly effective sorbents for water purification from various harmful substances of technogenic origin is of certain scientific and practical interest since it allows solving the problems of more rational use of water resources in production processes and contributes to improving the environmental situation. The efficiency of the process depends on the physicochemical properties of the sorbent, such as bulk density, specific surface area, pore volume, and mechanical strength. At the same time, the nature of the desorbing agent, hardware design, preliminary preparation of the sorbent (activation), and process conditions (pressure, temperature, contact time) also play an important role. Adsorption methods are also distinguished by the simplicity of hardware design and relatively low operating costs [1].

It is noteworthy that an interesting trend is observed for sorption methods, which consists of the possibility of involving some types of waste as a feedstock to obtain effective sorbents for the extraction of other types of waste.

Carbon sorbents can be made from various materials of natural and artificial origin, and the development of these areas allows the improvement of the quality indicators of various technological processes. Carbon sorbents have a wide range of applications and, depending on their adsorption properties, can be used to purify water from dyes, phenols, ions of various metals, as well as to purify gases.

Large sources of pollution with various dyes, phenolic compounds, various metals, and gases are coal, petrochemical, and metallurgical enterprises, producing paper, food products, plastic products, rubber products, coloring substances, etc. [2,3].

This class of substances poses a serious threat due to the toxic and carcinogenic properties that accumulate in the body, leading to serious health consequences.

Given the large volumes of wastewater generated in the above-mentioned industries, the development of simple methods for obtaining high-performance sorbents from available raw materials is an urgent task.

Natural materials of plant origin are promising raw materials for the production of carbon sorbents that exhibit high sorption capacity to various kinds of pollutants. For example, the physicochemical activation of oil palm fiber leads to the formation of activated carbon, which has a high specific surface area (1354 m^2^/g), and the sorption capacity is not inferior to commercial activated carbons when cleaning aqueous solutions from methylene blue [4]. Activated carbon obtained from coke fines is superior in adsorption capacity to commercial activated carbon during adsorption purification of aqueous solutions from methylene blue [5].

Activated carbons obtained by impregnating durian seeds in KOH demonstrate a high sorption capacity to the Remazol brilliant blue reactive (RBBR) dye and provide a degree of purification of an aqueous solution of about 95% [6]. Also, a high degree of purification (98.53%) of Reactive Yellow 145 (RY 145) from the wastewater of the textile industry is observed when using a carbon sorbent obtained by activating teff straw with a solution of phosphoric acid [7].

The development of methods for obtaining effective carbon sorbents is also of interest for their application in the extraction of inorganic substances from aqueous solutions, such as metal ions, which, along with organic substances, pose a serious environmental problem. Pyrolysis of various materials of organic origin makes it possible to obtain carbon sorbents with a high absorption capacity of metal ions. Biochar obtained by pyrolysis of sewage sludge at 700 °C demonstrates good sorption capacity for Cu^2+^ and Zn^2+^ [8]. Bio-coal obtained by pyrolysis of avocado seeds and grapefruit peel in a 300 °C tubular furnace is also a promising raw material for the production of sorbents for the removal of lead ions from aqueous solutions since their removal efficiency reaches 99% [9]. The high efficiency of lead removal from solutions is shown by biocarbon adsorbents obtained by pyrolysis of wood biomass, providing complete removal of lead [10].

One of the important advantages of activated carbons is also the possibility of improving their properties by modifying them with particles of various metals. This makes it possible to obtain sorbents for the extraction of certain types of substances. The efficiency of removing chromium ions from wastewater reaches 95% when using activated carbon-containing zero-valent iron [11]. A cellulose-based carbon sorbent obtained from wastepaper and modified with iron oxide and graphene oxide particles can be used in solid-phase extraction of antibiotics since it has sorption properties no worse than analogues [12]. Pyrolysis of a mixture of brown coal with poplar leaves in a ratio of 3:2 in a nitrogen atmosphere and subsequent activation modification of the resulting coal impregnated with solutions of zinc chloride and iron chloride in a fixed-layer reactor makes it possible to obtain a carbon sorbent with a high adsorption ability to remove lead and cadmium ions [13].

The main advantage of the above methods of obtaining carbon sorbents from materials of plant origin is the availability of raw materials for the preparation of sorbents and relatively low costs. However, the presence of such raw materials is typical for individual regions of the planet, and therefore the search for raw materials for carbon sorbents is interconnected, taking into account the peculiarities of geographical location with the corresponding resources. For countries with limited plant resources, coal reserves, coal, and waste from the coal industry are of interest for the production of sorbents. Brown coal can be considered a raw material for the production of industrial sorbents. For example, a fine fraction of brown coal 212–425 µm without preliminary physicochemical treatment exhibits a high adsorption capacity to methylene blue and surpasses activated carbons obtained from waste in this indicator [14]. The assessment of the adsorption properties of some varieties of coals is carried out using an integrated approach [15]. The fraction of brown coal with a size of 1–2 mm without preliminary preparation has a high adsorption capacity to copper ions, having a smaller specific surface area than activated carbons, which makes this material attractive for use in wastewater treatment processes [16].

Waste from the coal industry, such as fine coal fraction, is an excellent raw material for the production of carbon sorbents with a structure with a developed pore system. With the help of chemical and physical treatment methods, activated carbons with a high specific surface value can be obtained from coals [17]. Carbon sorbents from coke fines can be used to extract gold ions from cyanide solutions, while the degree of gold extraction is 94% [18]. Activation of a fine fraction of sub-bituminous and highly volatile bituminous coal with an alkaline solution leads to the formation of carbon sorbents, which are not inferior to industrial sorbents in static capacity for phenol and chloroform [19]. Activated carbons obtained by alkaline treatment of highly volatile bituminous coals have a high adsorption capacity to CO_2_, which, with relatively low costs and availability of raw materials, is a promising method for obtaining sorbents for air purification [20]. When these carbons are modified with potassium hydroxide, their subsequent activation with an ammonia solution with the addition of sodium tetraborate decahydrate increases the microporosity and the specific surface area [21].

This paper presents studies on the production of carbon sorbents from coke fines of coking coal from the Rapid formation of the Karaganda coal basin (Republic of Kazakhstan). The initial raw material for activated carbon was a fine fraction of medium-temperature coke with a size of 2–5 mm activated in a muffle furnace when exposed to water vapor. The method allows the involvement of a by-product of coke chemical production—coke fines—in the production of carbon sorbents.

The reactor design allows the combining of the stages of carbonation and steam activation. The method is characterized by the simplicity of hardware execution, in which no preliminary preparation of the source material and the creation of a protective atmosphere is required. Earlier, the authors studied the possibility of obtaining a carbon sorbent from coke fines, and the studies conducted showed the prospects of this method [22].

## 2. Results and Discussion

For the resulting activated carbon, physicochemical characteristics such as the degree of combustion loss, bulk density, sorption capacity for phenol, iodine and methylene blue, and the total pore volume depending on the duration of the activation process at a temperature of 850 °C were also established. Physicochemical characteristics are presented in Table 1.

Degree of combustion loss, also known as carbon burn-off or combustion yield, refers to the amount of carbonaceous material that is lost or consumed during the combustion process. It represents the incomplete combustion of carbonaceous substances, resulting in the formation of carbon dioxide (CO_2_) and other combustion byproducts. The degree of combustion loss can be influenced by various factors, including the combustion conditions, the nature of the carbonaceous material, and the presence of any impurities or inhibiting factors. High temperatures, sufficient oxygen supply, and longer combustion times generally promote more complete combustion and lower combustion losses. Incomplete combustion can occur due to insufficient oxygen supply, low combustion temperatures, or the presence of impurities that interfere with the combustion process. In such cases, not all of the carbonaceous material is converted to carbon dioxide, resulting in the formation of carbon monoxide (CO) or elemental carbon (soot) and potentially other harmful byproducts.

The degree of combustion loss is important to consider, especially in industrial processes and energy production, as incomplete combustion can lead to lower energy efficiency, increased emissions of pollutants, and the formation of particulate matter. To minimize combustion losses and promote more complete combustion, it is crucial to optimize combustion conditions, ensure sufficient oxygen supply, and control the presence of impurities or inhibiting factors.

It was found that the sample obtained at 120 min of activation has a maximum adsorption capacity for methylene blue and total pore volume compared to samples obtained at 60 and 90 min of activation (Table 1). The difference in the sorption capacity of iodine for all three samples was insignificant. Since the sample obtained at 120 min of activation showed the best results, it was taken for further study. An increase in the activation time >120 min led to an increase in the degree of combustion loss, and the maximum adsorption capacity for phenol and methylene blue practically did not change.

Chromatograms and concentrations of gaseous products at 60, 90, and 120 min are presented in Figure 1 and Table 2.

Gas extraction from the gas outlet showed that when coke fines are activated at the experimental installation, the exhaust gases contain H_2_, CO_2_, CH_4_, and CO.

The formation of methane may be associated with the decomposition of some of the organic components as a result of thermal exposure, which can also be observed by the change in the hydrogen content in the feedstock and the resulting activated carbon.

The main physicochemical characteristics of the coal activated at 120 min and the initial coke fines were compared to assess the changes. Comparative characteristics are presented in Table 3

The resulting activated carbon, unlike the starting material (coke fines), has a high degree of carbonation, there are no volatiles and phosphorus, and the sulfur and moisture content decreases (Table 3). At the same time, you can notice a noticeable decrease in the hydrogen content (Table 3), which takes up more than half of the exhaust gases during activation (Table 2).

SEM image of the surface of the coke breeze, presented in Figure 2, shows the presence of a small porosity.

Scanning electron microscopy (SEM) image showed that a sample of activated carbon obtained at 120 min has a more porous structure than coke fines. SEM image is shown in Figure 3.

By the method of low-temperature nitrogen adsorption, it was found that after the activation of coke fines by water vapor, a significant increase in the pore volume occurs. Adsorption isotherms are shown in Figure 4. At the same time, in the relative pressure region P/P_0_ = 0.2, there is a pronounced transition of monolayer adsorption into multilayer. Starting from P/P_0_ = 0.8, there is a transition of the curve to a larger slope, which is associated with the presence of wide meso- and macropores in the activated carbon.

In general, the nitrogen adsorption isotherm curve obtained for activated carbon resembles an isotherm of the second type [23].

Analysis of the pore size distribution showed that steam activation of coke fines led to a significant increase in the size distribution of meso- and macropores. Curves of the pore size distribution are presented in Figure 5.

With an increase in the degree of carbonation of coal, the pore size increases and the development of mesopores and macropores occurs. This can lead to an increase in the total volume of pores, which improves the adsorption properties and increases the capacity of coal to retain various substances. An increase in the degree of carbonation can increase the adsorption capacity of coal by increasing the volume of pores and surface, which allows coal to retain more substances. However, the selectivity of adsorption may vary depending on the type of adsorbate and its interaction with the surface of coal. It is important to note that the optimal degree of carbonation of activated carbon may depend on the specific application or process for which it will be used. Different degrees of carbonation may be preferred for different types of adsorbents.

According to the results of the analysis of the specific surface by the BET method, it was found that the average value of the specific surface of the initial coke fines was 38 ± 0.3 m^2^/g and 301 ± 10.8 m^2^/g for the coal obtained after activation.

As a result of FTIR spectroscopy, almost identical values of the absorption bands of bond oscillations were obtained for coke fines and activated carbon.

On the FTIR spectra shown in Figure 6, the differences in values are insignificant, and the differences are mainly in the absorption intensity. For activated carbon, the intensity of the absorption bands is much higher.

In the coke fines and the resulting activated carbon, the absorption bands at 621.15 cm^−1^, 775.45 cm^−1^, and 871.93 cm^−1^ indicate the presence of fluctuations in the out-of-plane bonds of C-H benzene derivatives. The presence of fluctuations in the C-O bond of esters in coke fines is indicated by the absorption band at 983.82 cm^−1^ and 987.67 cm^−1^ in activated carbon. The absorption bands at 1107.28 cm^−1^ and at 1114.99 cm^−1^ for activated carbon correspond to the fluctuations of the C-O bond of esters in coke fines. There may also be fluctuations in the C-O lactone bond in coke fines at 1400.5 cm^−1^ and 1361.91 cm^−1^ in activated carbon. Fluctuations of C=C bonds in polyaromatic hydrocarbons contained in coke fines and activated carbon at 1604.97 cm^−1^ have been established. Further, fluctuations in the C=O bond of lactones, ketones, and carboxylic anhydrides are observed in coke fines and activated carbon at an absorption band of 2361.16 cm^−1^. The presence of fluctuations in the C-H bonds of aliphatic hydrocarbon bonds is indicated by the absorption band 2831.85 cm^−1^. There are also absorption bands corresponding to the fluctuations of the R-OH bond in hydroxyl, carboxyl, and phenolic compounds present in coke fines, determined by the absorption bands 3437.57 cm^−1^ and 3734.64 cm^−1^ and at 3445.28 cm^−1^ and 3738.5 cm^−1^ for activated carbon.

The maximum adsorption capacity for phenol was 74.94 mg/g, and it was 145 mg/g for methylene blue. Table 4 shows the comparative characteristics of the activated carbon obtained by the authors of this article and activated carbons obtained by other researchers.

The maximum adsorption capacity of the resulting activated carbon to phenol and methylene blue may be due to the fact that, compared with the starting material, the resulting activated carbon has more intense absorption bands of oxygen-containing functional groups (hydroxyl, carboxyl, phenolic). These groups have an effect on the ion exchange capacity of coals, which in turn affects the adsorption properties [29].

Functional groups on the surface of activated carbon can significantly affect the sorption process of various substances, including phenol and methylene blue. Here are some examples of functional groups and their effect on sorption:

Hydroxyl Groups (-OH): Hydroxyl groups have a positive charge and can interact with anions or polar molecules. They can form hydrogen bonds with phenol and methylene blue, which contributes to their retention on the surface of coal.

Carboxyl Groups (-COOH): Carboxyl groups can also form hydrogen bonds and interact with anions or polar molecules. They are capable of forming ion-exchange bonds with positively charged parts of phenol and methylene blue.

Amino groups (-NH_2_): Amino groups can form bonds with acidic groups in phenol and methylene blue, such as hydroxyl and carboxyl groups. This can contribute to their retention on the surface of coal. Aromatic Structures: Activated carbon may contain aromatic carbon structures that can interact with the aromatic parts of phenol and methylene blue by van der Waals attraction forces. Anion exchange groups: Some functional groups on the surface of activated carbon can serve as anion exchange groups capable of forming ion-exchange bonds with anions, such as phosphates or sulfates in methylene blue. The sorption efficiency and the effect of functional groups may vary depending on specific conditions, such as the pH of the medium, temperature, and concentration of the substance. The complex interaction of various functional groups and their influence on sorption requires further study in each case.

The use of the resulting activated carbon for the purification of tap water leads to an improvement in some of its qualitative indicators. Characteristics of tap water before and after purification with the obtained activated carbon are presented in Table 5.

Table 5 shows that after water purification using obtained activated carbon, there is a decrease in the calcium content by more than two times, as well as a significant decrease in the content of sulfates and bicarbonate. There is also a decrease in total mineralization.

A change in the content of various elements in tap water during purification with the resulting activated carbon was also found. The content of various elements in tap water during purification with the obtained activated carbon is shown in Table 6.

The analysis showed that the resulting activated carbon also helps to reduce the content of compounds such as barium, boron, iron, sulfur, and strontium.

## 3. Materials and Methods

### 3.1. Preparation of Raw Materials for the Production of Activated Carbon

The initial raw material for the production of activated carbon was taken from the screening of coke production of coking coal of the Rapid formation of the Karaganda coal basin, which consists of coke fines of class 0–10 mm. This material was crushed on a jaw crusher and sieved on a vibrating stand to obtain a fraction with a size of 2–5 mm.

### 3.2. Obtaining Activated Carbon from Coke Fines

The installation for producing activated carbon from coke fines is a muffle furnace PM-14M1-1200 (LLC “EVS”, Russia), which inside there is an activation reactor made of heat-resistant stainless steel. The reactor is equipped with a tray with a lattice bottom with a hole size of 1×1 mm and a type K thermocouple for temperature measurement and has two pipelines: one for supplying water vapor and the second for removing gases.

Heating of the furnace is carried out by means of heaters built into ceramic muffle plates, which form the working chamber of the furnace and ensure uniform temperature distribution throughout the chamber volume. The heating is controlled by a microprocessor controller.

The scheme for obtaining activated carbon from coke fines is shown in Figure 7.

Before the activation process, coke fines were dried at 120 °C for 3 h. Activation of crushed coke fines (2–5 mm) in an amount of 2 kg was carried out at a temperature of 850 °C. When this temperature was reached, water vapor was supplied to the reactor from the steam generator for 120 min. A series of experiments with different durations of the process was carried out: 60, 90, and 120 min. After the activation and cooling process of the plant was completed, the yield of the resulting sorbent, the degree of combustion loss, and bulk density were determined.

### 3.3. Determination of Gas Composition

The analysis of gaseous products was carried out on a Crystallux 4000 M chromatograph (NPF Meta-Chrome, Russia) with a 2DTP/PID detector module; on a 3 m NaX column, d-3 mm for permanent gases; and a 3 m Porapak R column, d-3 mm for hydrocarbon gases and carbon dioxide. Quantitative calculation of chromatographic information was carried out using the program “NetChrom V 2.1”. The accuracy of the measurement results was evaluated, taking into account the calculations of the standard deviation. The value of the correction coefficient performed on 5 measurements was ~2.78 with a 95% confidence interval.

### 3.4. Determination of Adsorption Ability of Activated Carbon

Determination of the ability to remove phenol was carried out similarly to the method described in the work of E. Lorenc-Grabowska et al. [30]. The weight of the resulting activated carbon 0.05–0.2 g was placed in a conical Erlenmeyer flask, and 100 cm^3^ of an aqueous solution with a phenol content of 150 mg/L was poured. These mixtures were subjected to shaking in a thermoshaker at a temperature of 25 °C. Then, after filtering, the optical density of the solutions was measured using a LEKI SS2107 spectrophotometer at a wavelength of 270 nm. The value of adsorption (absorption capacity (*A*), mg/g) was determined by the formula:(1)$$A=\frac{(C_{0}-C)V}{m}$$
where *C*_0_ is the initial concentration of phenol in solution (mg/L); *C* is the final concentration of phenol in solution (mg/L); *V* is the volume of solution (L); *m* is the weight of the sample (g).

The adsorption capacity of methylene blue was determined as well as phenol.

### 3.5. Electron Microscopy

The morphology of the surface of the resulting activated carbon was studied using a scanning electron microscope TESCAN MIRA 3 LMU (TESCAN, Brno, Czech Republic) at an accelerating voltage of 20 kV.

### 3.6. Determination of the Specific Surface of Activated Carbon

The specific surface area of the activated carbon obtained was determined using low-temperature nitrogen adsorption by the BET method on the Sorbi MS device (Novosibirsk, Russia).

### 3.7. IR Spectrometry of Activated Carbon

The preparation of samples for IR spectrometry was carried out by pressing tablets, and pure KBr, previously crushed and dried, was used as the matrix substance.

The tablets were pressed in the following sequence: 2 mg of the test substance was weighed on analytical scales, and KBr powder was added, bringing the total weight of the sample to 300 mg. The prepared suspension was placed in an agate mortar for crushing and mixing. The resulting mixture of the test substance and KBr was evenly poured into a mold and pressed at 78.45 kN, obtaining a tablet with a diameter of 13 mm with a thickness of about 1 mm.

IR spectra were recorded on the FSM-1201 device (Infraspec LLC, RF) in the Fspec program (ver. 4.0.0.2) in the range of 400–4000 cm^−1^ in the transmission mode (with resolution of 8.0 cm^−1^).

## 4. Conclusions

The physicochemical chemical properties of activated carbon obtained from coke fines have been studied. It has been established that thermal steam activation of this material up to 850 °C with a process duration of 120 min is optimal for obtaining activated carbon with a developed surface. The average value of the specific surface area increases by about eight times compared to the raw material, while the resulting activated carbon has mainly a meso- and macroporous structure. The conducted studies have shown that this method allows activated carbons to be obtained from a low-liquid fraction (undersized coke) coke chemical production. The resulting activated carbons exhibit a high sorption ability to remove phenol (74.94 mg/g) and methylene blue (145 mg/g) from aqueous solutions, as well as some chemical elements from tap water. Purification of tap water with the resulting activated carbon leads to a decrease in the content of some chemical elements and a decrease in hardness.

## Figures and Tables

**Figure 1 molecules-28-05661-f001:**
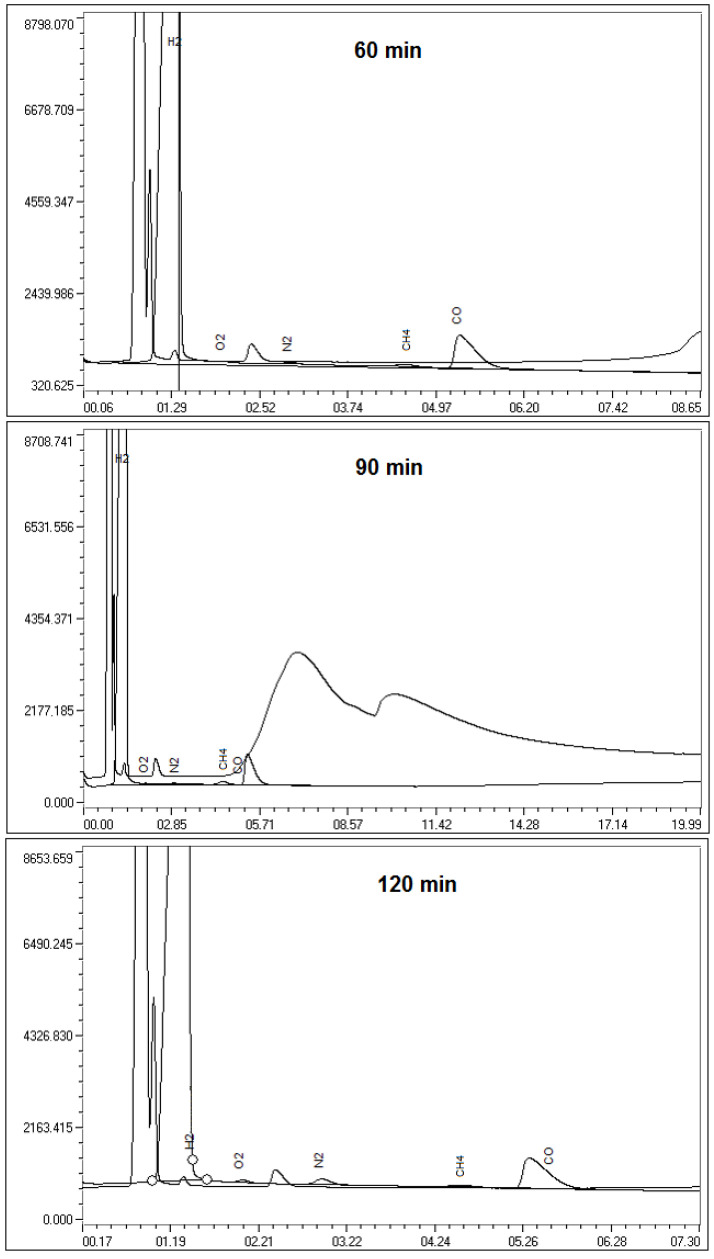
Chromatograms of gaseous products at 60, 90, and 120 min.

**Figure 2 molecules-28-05661-f002:**
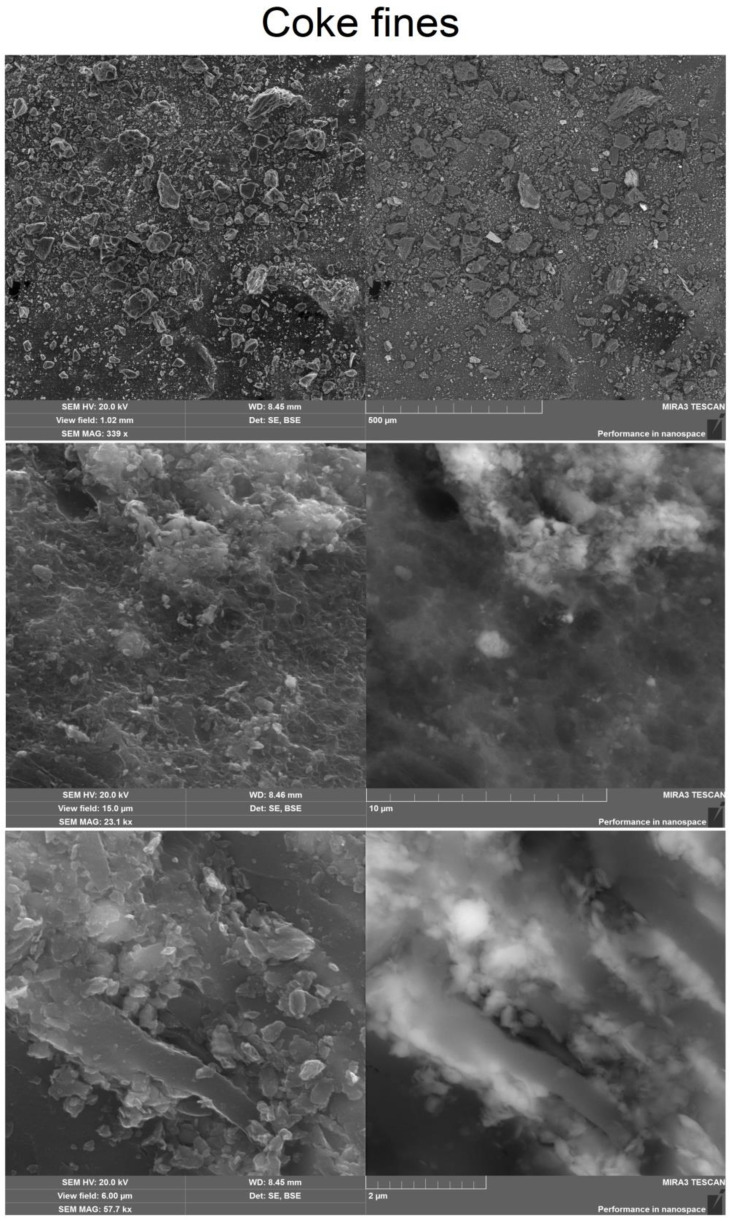
SEM image of coke fines.

**Figure 3 molecules-28-05661-f003:**
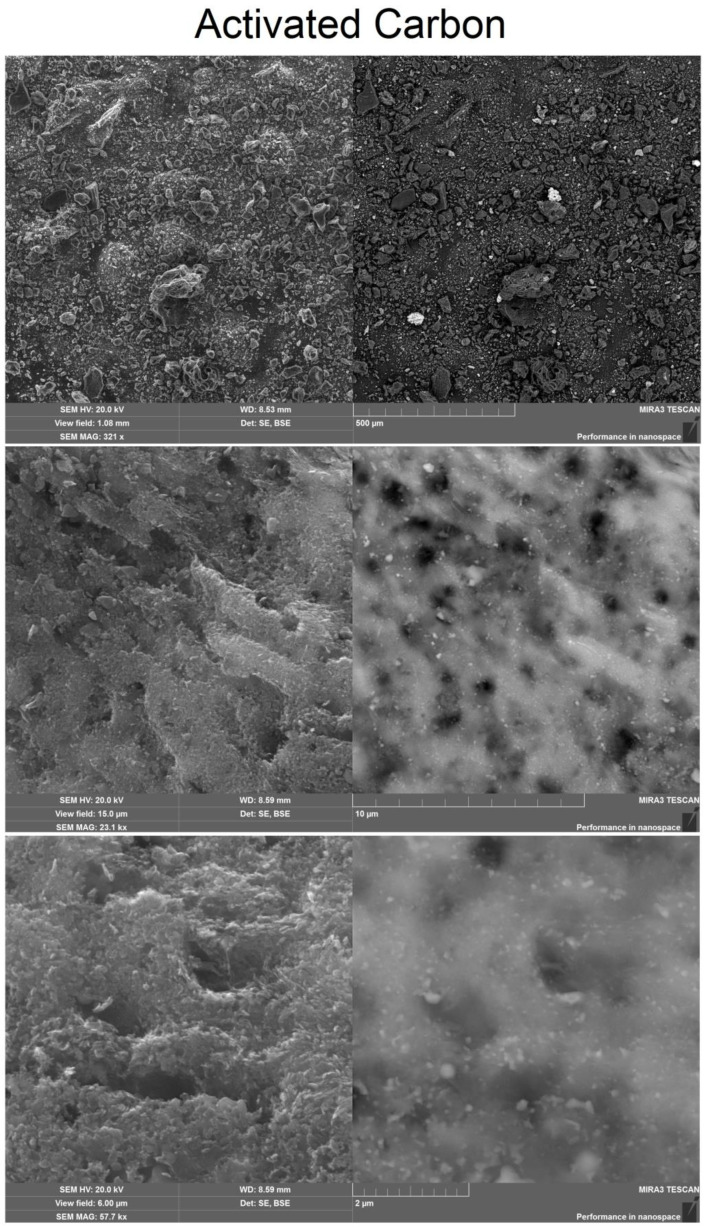
SEM image of activated carbon.

**Figure 4 molecules-28-05661-f004:**
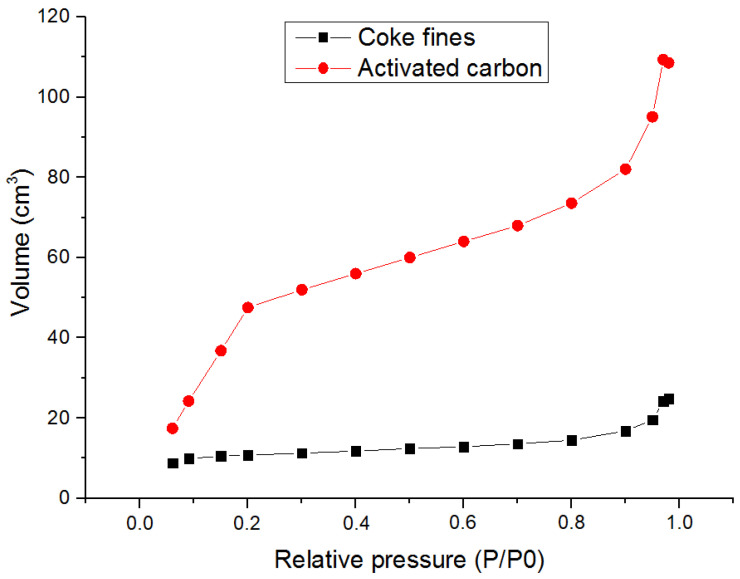
Adsorption isotherm of coal fines and activated carbon.

**Figure 5 molecules-28-05661-f005:**
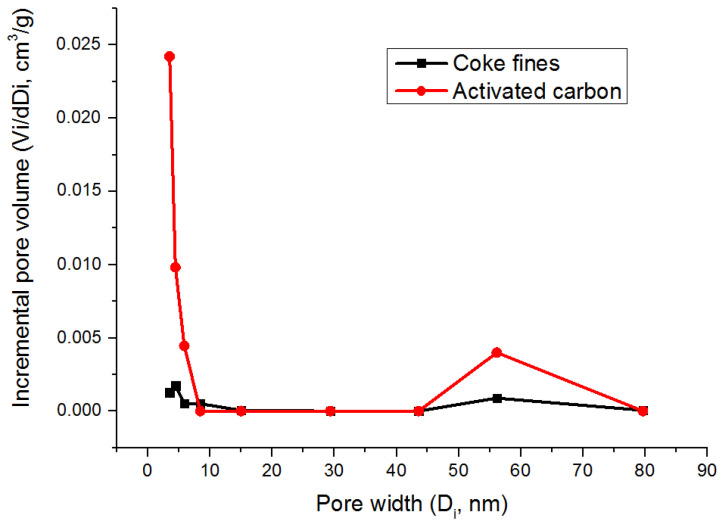
Pore size distribution of coke fines and activated carbon.

**Figure 6 molecules-28-05661-f006:**
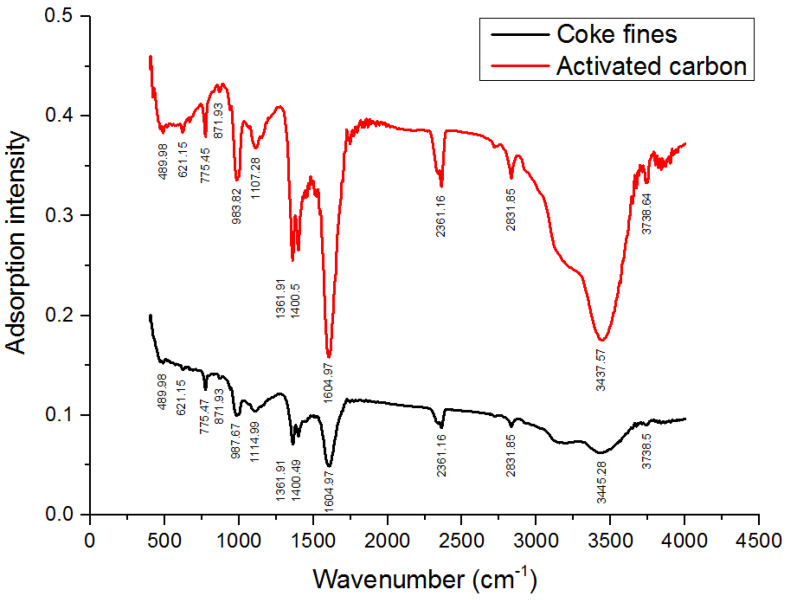
FTIR spectra of coal fines and activated carbon.

**Figure 7 molecules-28-05661-f007:**
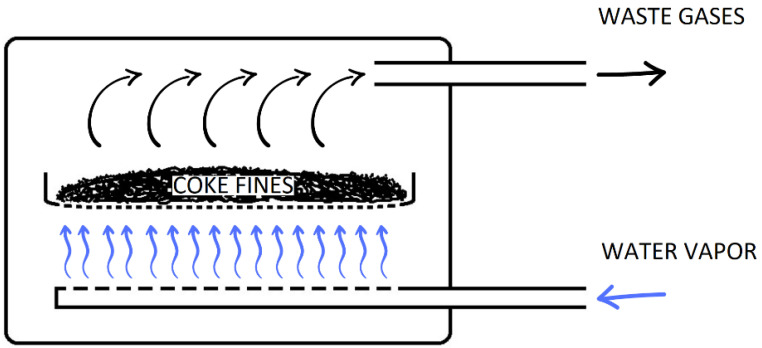
The scheme for obtaining activated carbon from coke fines.

**Table 1 molecules-28-05661-t001:** Dependence of the physicochemical characteristics of the activated carbon obtained on the duration of the process.

Index	Duration of the Process, min		
	60	90	120
Degree of combustion loss, %	11.97	27.90	29.83
Bulk density, g/L	437	417	395
Sorption capacity for iodine, %	23.43	25.87	26.75
Sorption capacity for methylene blue, mg/g	71	93	145
Total pore volume by water, cm^3^/g	0.49	0.54	0.75

**Table 2 molecules-28-05661-t002:** The composition of the exhaust gas during the activation of coke fines at a temperature of *t* = 850 °C and at different durations of the process.

Time, min; *t* = 850 °C	Gas Concentration, %					
	H_2_	CO_2_	CH_4_	CO	O_2_	N_2_
60	54.18 ± 0.03	10.33 ± 0.03	0.9 ± 0.02	33.21 ± 0.01	0.2 ± 0.01	1.36 ± 0.04
90	57.45 ± 0.02	11.68 ± 0.05	1.13 ± 0.02	32 ± 1.08	0.13 ± 0.02	0.84 ± 0.02
120	53.56 ± 0.04	10.53 ± 0.02	0.78 ± 0.03	30.82 ± 0.03	0.63 ± 0.05	3.69 ± 0.05

**Table 3 molecules-28-05661-t003:** Physicochemical characteristics of coke fines and the resulting activated carbon.

Index	Coke Fines	Activated Carbon
Carbon content per dry mass, C^d^, %	62.6	75.29
Hydrogen content per dry mass, H^d^, %	18.2	3.93
Ash content per dry mass, A^d^, %	29.0	25.24
Volatile substance yield per dry ash free mass, V^daf^, %	4.0	-
Phosphorus content P, %	0.02	-
Mass fraction of sulfur S, %	0.49	0.42
Mass fraction of total moisture in working condition, %	12	2

**Table 4 molecules-28-05661-t004:** Surface areas and maximum adsorption capacities of phenol and methylene blue for activated carbon obtained in the present work and by other researchers.

Activated Carbon (AC)	Surface Area (BET), m^2^/g	Phenol Adsorption, mg/g	Methylene Blue Adsorption, mg/g
AC present study	301	74.94	145
FNAC [24]	2869	76.26	125.16
Sugarcane bagasse AC [25]	>1000	159	148.8
HGAC [26]	619.2	197.11	175.02
vet-H_2_O [27]	1185	145	375
D-AC500–800 [28]	1951	549.6	841.93

**Table 5 molecules-28-05661-t005:** Characteristics of tap water before and after purification with the obtained activated carbon.

Index	Before Purification	After Purification
**Cations**, mg/L		
Sodium and potassium	140 ± 3	138 ± 2
Calcium	52 ± 3	20 ± 2
Magnesium	27 ± 2	24 ± 2
Fe	-	-
Hydrogen ion	-	-
**Total**	219	182
**Anions**, mg/L		
Chloride	177 ± 3	156 ± 2
Sulphate	125 ± 2	77 ± 2
Bicarbonate	201 ± 3	73 ± 3
Carbonate	-	54 ± 2
Nitrate	<0.3 ± 0.2	<0.3 ± 0.1
**Total**	504	360
Total hardness	4.80 ± 0.01	3,00 ± 0.01
Carbonate hardness	3.30 ± 0.02	3,00 ± 0.01
pH	8.0 ± 0.14	9,5 ± 0.01
Dry residue, mg/L	622 ± 1	506 ± 4
Total mineralization, mg/L	723 ± 2	542 ± 1

**Table 6 molecules-28-05661-t006:** The content of various elements in tap water during purification with the obtained activated carbon.

No.	Elements	Before Purification	After Purification
		mg/L	mg/L
1	Aluminium	0.081	0.079
2	Barium	0.0800	0.0396
3	Beryllium	<0.00010	<0.00010
4	Boron	0.252	0.103
5	Bismuth	<0.010	<0.010
6	Tungsten	<0.010	<0.010
7	Iron	0.254	0.124
8	Cadmium	0.00031	0.00022
9	Calcium	44.6280	19.1190
10	Cobalt	0.0018	0.0016
11	Magnesium	27.683	23.668
12	Manganese	0.0024	0.0013
13	Copper	0.0053	0.0024
14	Arsenic	<0.0050	<0.0050
15	Sodium	119.07	117.85
16	Nickel	0.0067	0.0034
17	Tin	<0.0050	<0.0050
18	Lead	0.0024	<0.0010
19	Selenium	<0.0050	<0.0050
20	Sulfur	51.086	37.364
21	Silver	<0.0050	<0.0050
22	Strontium	0.7557	0.4981
23	Antimony	<0.0050	<0.0050
24	Thallium	<0.0050	<0.0050
25	Titanium	0.0011	0.0010
26	Zinc	0.0150	0.0150

## Data Availability

Data are contained within the article.

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
