# Peer review of "Production of Activated Carbon from Sifted Coke and Determination of Its Physicochemical Characteristics"

_molecules, 2023, doi:10.3390/molecules28155661_

Round 1

Reviewer 1 Report

In this study, the coal sorbents from coke fines have been obtained. The coke fines were activated at 850 °C with steam treatment. The influence of the duration of the activation process (60, 90 and 120 minutes) was studied. The adsorption capacity to remove methylene blue and phenol from aqueous solutions was surveyed. Obtained activated adsorbent was studied by scanning electron microscopy, low-temperature nitrogen adsorption (BET) and FTIR spectroscopy.

To be published in Journal of Molecules, this manuscript needs major revision.

Please see the specific comments below:

- The results of the manuscript are limited: The investigation of activation of coke fine is limited. Only one factor, activation time (60, 90 and 120 minutes), was investigated; other important factors are not considered. Furthermore, it is unreasonable to limit the investigation of activation time to no more than 120 min because the research results show that the carbon sample activated for 120 minutes is the best, so it is necessary to increase the activation time. The methods for analyzing the physic-chemical and structural properties of the materials are limited with scanning electron microscopy, low-temperature nitrogen adsorption (BET) and FTIR spectroscopy. Then, the processes taking place during the activation cannot be explained. There are no comments for "Table 3. Physicochemical characteristics of coke fines and the resulting activated carbon" (page 5).

In general, the manuscript is presented in the form of a report of results, without comments and explanations of the results, so the scientific soundness is poor

- “Determination of the brightening ability by methylene blue” and “Assessment of the ability to remove phenol from aqueous solutions” should be performed according to standard methods and cited the reference.

- English must be extensively edited, the grammar and typographical errors mush be checked and corrected.

English must be extensively edited, the grammar and typographical errors mush be checked and corrected.

Author Response

The article has been corrected in accordance with the comments.

Reviewer 2 Report

The weak points cited below:

In the abstract, more results should be added.

In the Introduction, the novelty is still limited in the Introduction, and the author should state the previous studies on the use of carbon sorbents from fines of coking coal, to give the comparison with the present research.

The part of Materials and Methods should be placed in front of the Results and Discussion. The quality of the figures must be improved, such as Figure 1, Figure 3, Figure 4 and Figure 5. The standard deviation of the data before and after the purification should be provided.

In the Materials and Methods, was the nitrogen protection used during the preparing of activated carbon from coke fines. How to measure the concentrations of methylene blue and phenol.

In the Results and Discussion, the discussion is too weak, just the statement of data, please enhance the discussion, including the comparing with previous studies.

In the conclusion, the statement of “significant increase”, “in can be assumed that” should be avoided.

More references should be added to give comparisons with present results.

Should be checked carefully.

Author Response

(The authors gave the same response as above.)

Author Response

(The authors gave the same response as above.)

Reviewer 4 Report

The manuscript in its current state is not suitable for publication due to the lack of significant results.

The article presents selected measurement results without referencing the results of other authors. In particular, the results of the performance of the obtained material as an adsorbent are not compared to other materials, including the frequently mentioned 'industrial KAD-iodine'. Additionally, it is not stated whether the activation method is a standard method for obtaining activated carbon. The Conclusions section does not provide values for the obtained parameters.

In the Introduction, a number of sources regarding the obtaining of activated carbon are provided, but none of them are used in the Discussion section. It appears that they are only included to inflate the article's volume.
Lines 103-118 could be removed as they do not contain information relevant to the conducted research.
In lines 149-158, the aim of the study is not clearly stated, but the results are presented, which should be in Conclusions.

The paper contains numerous language errors and imprecise phrasings. I will not list them here, as I suspect that the paper will be rejected by other reviewers due to other shortcomings.
In Table 1, the last line - 'total pore volume by water m3/g = 0.49' - appears to be an inflated result.

Figure 1: All figures should have the same scale and be larger, as well as arranged vertically.
Table 2: No measurement errors are indicated.
Table 3: What do the letters 'd' and 'dat' in the superscript index signify? What does the parameter 'Structural strength' mean?
Figure 2: There are no or invisible descriptions of SEM images.
Figure 3: Points should be marked instead of a curve not resulting from any equation. The measurement temperature should also be provided. The caption lacks specificity as to whether the pressure P and Po is for air or nitrogen.
Line 215: 'increase in the distribution of' - 'the size distribution of' or 'the distribution number of meso- and macropores'?
Figure 4: The units should be checked.
Table 4: Information about the measurement method should be provided. Are the data for KAD-iodine from the manufacturer? If so, the source of this data should be provided.
Line 259, Tables 4 and 5: The notation of 'g/L', 'g/l', 'mg/L', and 'mg/l' should be unified.

Table 5 - an additional column of data for KAD-iodine adsorbent should be added to the table - otherwise the data has limited value for the reader. A column with the percentage "purification ratio" would be useful.
In Table 5, a word written in Cyrillic alphabet was left unchanged.
Table 6 - comments as for Table 5.
Line 309 - change "850 0C" to "850 °C".
Line 323 - what does "correction coefficient" mean in this sentence?
Line 335 - "conventionally accepted" - what does it mean, where is the scale of acceptability or the measurement standard for this quantity in the source?
Lines 341-344 - what was the volume of the solution passed through the column?
Line 245 - does "normal" mean exactly 1013.2 hPa, or just at atmospheric pressure?
Line 368 - what does "8.0 Tc" mean??? Is it 8 atm???
Line 371 - (wave-number 8.0 cm-1) should be changed to (with resolution of 8.0 cm-1).
Line 373 - "5. Conclusions" should be numbered as 4.
In References, four papers on phenol removal are listed - however, no data on phenol is provided in Results. Citing works 3, 9, 10 and 14 is unnecessary.
Line 425 - the name Zbigniew should be removed.

To increase the scientific and practical value of the article, activation should also be performed at other temperatures, e.g. 650, 750, 850, and 950 °C, with an extended time of 180 minutes.

I will comment on the revision if it happens.

Author Response

(The authors gave the same response as above.)

Round 2

Reviewer 1 Report

In the revised manuscript, some of the comments raised in the previous review have been modified/explained. Table 4 has been added to compare the results of the present study with previous studies. However, this table shows that the results obtained in the present manuscript are not superior to those previously published both in terms of specific surface area and adsorption capacity for phenol and methyl blue. Importantly, the key issues raised in the previous review have not been satisfactorily resolved. Specifically,

-   For Comment 1 in the new draft the paragraph “An increase in the activation time >120 minutes led to an increase in the degree of combustion loss and the  adsorption capacity for phenol and methylene blue practically did not change.” (Lines 242-243) was added. However, no results were shown to prove it.

-       For Comment 2: “The methods for analyzing the physic-chemical and structural properties of the materials are limited with scanning electron microscopy, low-temperature nitrogen adsorption (BET) and FTIR spectroscopy. Then, the processes taking place during the activation cannot be explained.” has not been modified, supplemented, and almost not improved.

-    For Comment 3: the comments and explanations of the results has not been improved. Then, the scientific soundness is still limited.

-       In the revised draft, there are still exist errors, for example:

+ size of 0-10 mm (line 10, Abstract, line 137)

+ In section “Determination of the brightening ability by methylene blue (176)” why describe phenol adsorption method? Why is phenol adsorption investigated by 2 methods (page 4-5, lines 176-200), so result of Phenol adsorption in Table 4 is from which method? Methyl Blue adsorption method and MB analysis method have not been described.

Given the above disadvantages, the article is not recommended to be published in the Journal of Molecules.

- Could be improved.

Author Response

the article has been corrected in accordance with the comments

Reviewer 2 Report

The authors have revised the manuscript according to my comments, so I recommend to accept the paper to be published in the Molecules.

Author Response

(The authors gave the same response as above.)

Reviewer 4 Report

Production of activated carbon from sifted coke and determination of its physicochemical characteristics  by A. T. Ordabaeva et al.

Despite encouragement from the reviewer, the authors decided not to compare the produced absorbent to 'industrial KAD-iodine,' which reduced the scientific value of the study. In its current state, it is a technical report of limited utility to other researchers. Even when limiting the study to only one input material, the authors could have compared the properties of the produced product for different material fractions and answer the question: Is crushing and sieving necessary step in production?

The authors have made several amendments and changes to the article and should carefully review the new sections.

Figures are wrongly numbered: First figure has number 6, Figure 1 should be Figure 2 etc in Lines 153, 252, 278, 290, 298, 327

Line 157 and 267 should say "coke fines" instead of "coconut fines."

Lines 179-182 “The weight of the resulting activated carbon is 0.05-0.2 g was placed in a conical Erlenmeyer flask, 100 cm3 of an aqueous solution with a phenol content of 150 mg/L was poured and subjected to shaking in a thermoshaker at a temperature of 25 °C.  break to 2 sentences.

Line 179 -  change  mg/l  to mg/L

Line 222 – “pressed at 8.0 tf (tonne-force), obtaining” change to SI units.

Line 229 – correct this “physicochemical characteristics as the Degree of combustion loss,” – no capital D in “Degree” and explain what it means

Line 238 – correct “120 minutes of activation has a maximum capacity for methylene blue ” to “120 minutes of activation has a maximum adsorption capacity for methylene blue”. Please do not mix adsorption with absorption with the whole paper.

Figure 1.  The same scales should be used for all figures.

Figure 2: There are no or barely visible descriptions of SEM images - white letters on the black background. The images have such low resolution that it is not possible to read the captions. Therefore, higher-resolution images should be included, or the captions should provide values of magnifications and other descriptions.

Table 4: The authors provide remarks [in square brackets] about the data sources for the description of the type of activated carbon, but these sources are not listed in the References.

Table 5: The authors added error values to the measured data, but they did not use the common notation:

Current: Sodium and potassium   140±3.08    138±2.12

Should be: Sodium and potassium 140±3       138±2

Or: Sodium and potassium        140.1±3.1   138.3±2.1

That means: In result and precision, the same number of decimal places should be provided.

Line 367 – change “the average: to :The average”

The authors have made several amendments and changes to the article and should carefully review the new sections.

Author Response

(The authors gave the same response as above.)

Round 3

Reviewer 4 Report

Review of Version 3 of the article "Production of activated carbon from sifted coke and determination of its physicochemical characteristics"

 The authors have corrected some errors and supplemented the text with explanations addressing the reviewer's concerns.

In their explanations in response to the review author's statements are repetitive and lack substantial substance or convincing arguments. The authors state, "The choice of certain fractions of coke fines is based on experience and research, which allow us to determine the optimal parameters to achieve the required characteristics of activated carbon." However, they do not provide any published work describing the research or optimization criteria.

Similarly, they write, "Fractions with certain pore characteristics are selected, which provide the desired ratio between the surface and the volume of the pores." However, they do not explain the basis on which finer fractions can have different pore characteristics than coarser fractions. They also do not specify the value of the "desired ratio." Moreover, it would be more appropriate to use "ratio between the volume and the surface," which is proportional to the linear dimension of the pores.

The authors state, "Fractions of coke fines are selected in such a way that the specific density is optimal for the activation process. This may be due to ensuring good gas permeability and efficient particle heating." However, gas permeability may be better for smaller fractions, and if we understand "efficient particle heating" as shorter heating time, it is a completely erroneous argument because the heating time for such fractions, as used by the authors, is on the order of seconds, while the process lasts for several hours. Therefore, it is a characteristic that can be completely disregarded.

I still believe that the authors should present more research indicating that the selected process parameters yield the best final product, meaning the product with the highest substance removal efficiency within a given time or the shortest substance removal time to a specific level.

In their response to the reviewer, the authors explained what "degree of combustion" means. However, the authors should provide the formula according to which this quantity is calculated in the article and put this formula in the paper.

Detailed remarks.

Line 58 - the authors mistakenly changed "coconut husk" to "coke fines" - they should restore "coconut husk" in that place as cited in reference [5].

Line 272 - it says "coal fines" - it should be "coke fines".

Lines 276-282 - The authors seem to have mistakenly used the word "coal" instead of "coke fines". Both terms refer to different materials. If they really meant "coal", then lines 276-285 are off-topic and should be removed.

In Table 4, the authors left verbal descriptions instead of inserting numbers used in References.

Conclusion: The presented material has little scientific value. It is merely a research report. The material would gain value if the authors at least added adsorption isotherms for different fractions of coke fines.